# Human Blood Serum Can Diminish EGFR-Targeted Inhibition of Squamous Carcinoma Cell Growth through Reactivation of MAPK and EGFR Pathways

**DOI:** 10.3390/cells12162022

**Published:** 2023-08-08

**Authors:** Dmitri Kamashev, Nina Shaban, Timofey Lebedev, Vladimir Prassolov, Maria Suntsova, Mikhail Raevskiy, Nurshat Gaifullin, Marina Sekacheva, Andrew Garazha, Elena Poddubskaya, Maksim Sorokin, Anton Buzdin

**Affiliations:** 1I.M. Sechenov First Moscow State Medical University, Moscow 119991, Russia; sorokin@oncobox.com; 2Shemyakin-Ovchinnikov Institute of Bioorganic Chemistry, Moscow 117997, Russia; shaban.na@phystech.edu (N.S.); buzdin@oncobox.com (A.B.); 3Moscow Institute of Physics and Technology, Dolgoprudny 141701, Russia; suntsova@oncobox.com; 4Engelhardt Institute of Molecular Biology, Moscow 119991, Russia; lebedevtd@gmail.com (T.L.); prassolov45@mail.ru (V.P.); 5World-Class Research Center “Digital Biodesign and Personalized Healthcare”, Sechenov First Moscow State Medical University, Moscow 119991, Russia; raevsky@oncobox.com (M.R.); podd-elena@yandex.ru (E.P.); 6Department of Pathology, Faculty of Medicine, Lomonosov Moscow State University, Moscow 119992, Russia; gaifulin@rambler.ru; 7Oncobox Ltd., Moscow 121205, Russia; garazha@oncobox.com; 8Omicsway Corp., Walnut, CA 91789, USA; 9PathoBiology Group, European Organization for Research and Treatment of Cancer (EORTC), 1200 Brussels, Belgium

**Keywords:** EGFR, HER-targeted cancer therapy, human blood serum, cetuximab, erlotinib, EGF, squamous cell carcinoma, drug resistance, ERK activity inhibition, MAPK molecular pathway

## Abstract

Regardless of the presence or absence of specific diagnostic mutations, many cancer patients fail to respond to EGFR-targeted therapeutics, and a personalized approach is needed to identify putative (non)responders. We found previously that human peripheral blood and EGF can modulate the activities of EGFR-specific drugs on inhibiting clonogenity in model EGFR-positive A431 squamous carcinoma cells. Here, we report that human serum can dramatically abolish the cell growth rate inhibition by EGFR-specific drugs cetuximab and erlotinib. We show that this phenomenon is linked with derepression of drug-induced G1S cell cycle transition arrest. Furthermore, A431 cell growth inhibition by cetuximab, erlotinib, and EGF correlates with a decreased activity of ERK1/2 proteins. In turn, the EGF- and human serum-mediated rescue of drug-treated A431 cells restores ERK1/2 activity in functional tests. RNA sequencing revealed 1271 and 1566 differentially expressed genes (DEGs) in the presence of cetuximab and erlotinib, respectively. Erlotinib- and cetuximab-specific DEGs significantly overlapped. Interestingly, the expression of 100% and 75% of these DEGs restores to the no-drug level when EGF or a mixed human serum sample, respectively, is added along with cetuximab. In the case of erlotinib, EGF and human serum restore the expression of 39% and 83% of DEGs, respectively. We further assessed differential molecular pathway activation levels and propose that EGF/human serum-mediated A431 resistance to EGFR drugs can be largely explained by reactivation of the MAPK signaling cascade.

## 1. Introduction

The human epidermal growth factor receptor (HER) family of receptors has four members: HER1-4, also called EGFR, neu, ErbB3, and ErbB4, respectively [1,2,3,4,5]. Several growth factors called HER ligands are known to be able to bind these receptors and activate them [6,7,8,9]. In their apo form, HER receptors exist in equilibrium between the monomeric and pre-dimerized states. After ligand binding to extracellular domains (ectodomains) of EGFR, HER3, or HER4, the holo receptors homo- or heterodimerize with proper assembly and become functionally active through reciprocal intramolecular tyrosine phosphorylation [10,11]. EGFR molecules are essential for the mediation of proliferative signals to cells: they regulate cell growth, survival, and differentiation via several signal transduction pathways [5,12,13,14,15]. Abnormal expression and dysregulated intracellular signaling through HER family members has a pivotal role in carcinogenesis, and hyperactivating mutations of EGFR have been identified in several cancer types [11,16,17,18,19,20,21].

A therapeutic approach has been developed to block EGFR receptor activities by targeted drugs. Monoclonal antibodies (mAbs) including cetuximab and panitumumab target extracellular domains of EGFR, and low molecular mass molecules including erlotinib and gefitinib inhibit the EGFR intracellular tyrosine kinase domain. The therapeutic use of EGFR inhibitors has been approved for the treatment of carcinoma cell subtypes of head and neck, colorectal, pancreatic, and lung cancers [22,23,24,25].

However, not all patients with tumors expressing high levels of EGFR respond to these drugs, and many cancers develop resistance to such treatments [26,27,28]. For example, even in the presence of activating mutations in EGFR, response rates to erlotinib and gefitinib are close to 50% [29]. Thus, the specific predictors of clinical response are of substantial practical significance. Several groups of putative response factors are discussed in the literature, including a proteomic spectra of blood serum samples [30], tumor mRNA expression biomarkers [31], transcriptome-based deduced activities of intracellular molecular pathways [32,33], EGFR expression level in tumors, and activating mutations of *EGFR* gene and of downstream regulatory kinases such as RAS family proteins and *BRAF* [34,35]. Efforts to discover and validate effective and clinically actionable biomarkers are ongoing to further refine treatment efficacy.

Certain indications suggest that EGFR and its ligands (epidermal growth factor (EGF), amphiregulin, HB-EGF, TGF-alpha) can be applied as serological biomarkers in relation to the prognosis and prediction of response to EGFR-targeted treatments in lung [36], ovarian [37], and colorectal cancer [38]. In cell culture studies, strong rescuing effects of HER ligands (EGF, neuregulin (NRG)) on cells treated with HER-targeted drugs were demonstrated [39,40,41,42]. For example, in cetuximab or gefitinib plus EGF treatments, less colon cancer cells could develop apoptosis [43].

Thus, tumor response can be affected by a variety of extracellular factors present in human peripheral blood. Concentrations of endogenous EGF and other HER ligands can vary in a patient’s body and, therefore, can influence treatment efficacy through the potential rescue of cancer cells during HER-targeted therapy [44]. Nonetheless, crosstalk between human serum and targeted drugs has not yet been sufficiently investigated.

Recently, we found that human peripheral blood modulates activities of HER-specific drugs cetuximab, erlotinib, and trastuzumab in cultured cells colony formation assay [42,44]. Here, we demonstrate that human serum suppresses the impact of EGFR-targeted drugs cetuximab and erlotinib on the growth rate of A431 squamous carcinoma cells. The A431 growth rate was measured in the presence of human peripheral blood serum samples taken from 23 human donors. Overall, the influence of human serum samples was essentially antagonistic for both erlotinib and cetuximab activities, and significant donor-to-donor variation was demonstrated. The anti-EGFR drugs inhibit cell cycle progression and cause G0/G1 phase arrest [45]. We demonstrate here that this G0/G1 arrest caused by cetuximab and erlotinib is abolished by human serum. Furthermore, the Ras/Raf/ERK signaling axis plays a crucial role in the growth and proliferation of tumor cells [46]. Specifically, ERK1/2 is a major regulator of cell survival and one of the main effector downstream targets of EGFR. In colorectal cancer cells, cetuximab induces cell death in an ERK-dependent manner [47,48], and enhanced ERK1/2 activity is a possible mechanism of cetuximab resistance in head and neck squamous cell carcinoma and colorectal cancer cells [49,50]. Hence, we measured the ERK1/2 kinase activity of treated cells. We found that in the presence of EGF or a mixed human serum sample, both drugs (cetuximab and erlotinib) essentially fail to inhibit ERK1/2 activity in A431 cells.

In total, RNA sequencing revealed 1271 and 1566 differentially expressed genes (DEGs) in the presence of cetuximab and erlotinib, respectively. Erlotinib- and cetuximab-specific DEGs significantly overlapped (*p* < 0.001 by permutation test). Interestingly, the expression of 100% and 75% of these DEGs restores to no-drug level when EGF or a mixed human serum sample, respectively, is added along with cetuximab. In the case of erlotinib, EGF and human serum restored the expression of 39% and 83% of DEGs, respectively. We further assessed molecular pathway activation levels and propose that EGF/human serum-mediated A431 resistance to EGFR drugs can be largely explained by reactivation of the MAPK signaling cascade.

## 2. Materials and Methods

### 2.1. Cell Culture

The squamous carcinoma cell line A431 (ATCC CRL-1555) was obtained from the collection of the Institute of Cytology, Saint-Petersburg, Russia. A431 cells were cultured at 37 °C and 5% CO_2_ in DMEM (Paneco, Russia) supplemented with 10% FBS (Biosera, South America) and a mixture of 2 mM L-glutamin, 4.5 g/L glucose, and 1% penicillin–streptomycin (Paneco, Russia).

### 2.2. Cell Grow Rate Measurement

The cells were plated on 24-well culture plates, ~5500 cells per well. The plates with an equal number of seeded cells were incubated for 24 h before treatment with EGF or EGFR-targeted drugs, or human sera. On the day the drugs were added, a control count was made to calculate the background value of cell growth. After 4 days, the medium was discarded, and treated cells were washed and trypsinized for 10 min. After resuspension, the cells were counted using a Neubauer Improved cell counting chamber. The cell growth (%) was calculated as a ratio of the number of cells minus the background value formed by treated and untreated cells (one control well was counted for five of treated wells). All experiments were conducted in at least three independent replicates.

### 2.3. Cell Cycle Analysis

The cells were plated on six-well culture plates at a density of 50,000 cells per well. Cells were incubated for 16 h, and drugs were added as indicated. Following 3–4 days of treatment, cells were detached with trypsin. Cells were then twice washed with PBS by centrifugation at 100× *g*. The cell pellet was fixed with ethanol (70%) at −20 °C for 30 min. The cells were centrifuged at 3800 rpm (1000× *g*) for 10 min; the pellet was washed with PBS and resuspended in 0.2 mL PBS containing 10 µg/mL of RNase AII and 2 µg/mL of propidium iodide. Cells were incubated at room temperature for 30 min and protected from light. The samples were analyzed by flow cytometry (FACS BD Accuri C6 Plus). All experiments were conducted in at least three independent replicates.

### 2.4. Cell Treatment for RNA Sequencing

Cell suspension was diluted to 27,000 cells per ml, and 5 mL aliquots of cell suspension were plated on 25 cm^2^ culture flasks. From the same cell suspension, aliquots with 0.5 mL per well were plated on wells of 6-well culture plates for the growth rate measurement. Cells were incubated for 16 h, and drugs were added as indicated. Examples: erlotinib, freshly diluted solution in growth media per flask to a final concentration of 800 nM; EGF, freshly diluted solution in growth media per flask to a final concentration of 2 ng/mL; 300 µL human blood serum per flask to a final concentration of 5.5%. To the control wells, 1/10 volume of the same drug solutions was added. Cells were treated with drugs in the flasks for 48 h before harvesting samples for RNA sequencing and for 4 days for cell counting. For RNA sequencing, the cells were detached with trypsin, washed with PBS, and cell pellets were stored at −70 °C in RNAlater solution before analysis. The human serum used for RNA sequencing profiling was a pool of sera obtained from eight donors (sd 2, 14, 15, 17, 18, 19, 20, 21) and mixed in equal volume concentrations.

### 2.5. EGFR-Targeted Drugs, EGF, and Human Serum Samples

Cetuximab (Erbitux), solution 5 mg/mL, was purchased from Merck and stored at 4 °C; erlotinib, dry powder, was purchased from Sigma-Aldrich and stored at −20 °C as 10 mM solution in DMSO. rhEGF, dry powder, was purchased from SCI store (Moscow, Russia) and stored at −20° C. Peripheral blood samples from unrelated healthy 23–64-year-old female donors were collected in two 8 mL Vacuette tubes containing pro-coagulant and gel (Greiner), and serum was prepared within 3–4 h upon blood collection: tubes were centrifuged at 2500 rpm for 15 min, sera were aliquoted and stored at −75 °C. For all the human biomaterials investigated, informed written consent to participate in this study and to communicate the results in the form of a scientific report was collected from the corresponding donors. The study was conducted in accordance with the Declaration of Helsinki; the consent procedure and the design of the study were approved by the ethical committee of the Vitamed Clinical Center, Moscow; date of approval: 6 October 2021.

### 2.6. In Vivo ERK Activity Assay

A431 cells with a stable expression of the ERK-KTR reporter were created by lentiviral transduction. Lentiviral particles were created by the cotransfection of HEK293T cells with pLentiCMV Puro DEST ERKKTRClover plasmid (Addgene, USA) [51], and plasmids encoding Rev, Gag/Pol, and VSV-G protein, as described in [52]. Lentiviral particles were used in a titer sufficient for 90% cell transduction. For nuclei segmentation, cells were stained by 500 ng/mL Hoechst-33342 30 min before imaging. Cells were imaged using a Leica DMI8 automated fluorescent microscope. For each biological repeat, 6 randomly selected fields were imaged. Hoechst-33342 was detected using a DAPI filter (EX:325–375, DC: 400, EM: 435–485), and mClover fluorescence was measured using an FITC filter (EX:460–500, DC: 505, EM: 512–542) with a 10x magnification lens. Illumination correction, nuclei and cytoplasm segmentation, and calculation of the ratio between the cytoplasm and nucleus was performed in CellProfiler 4.0.5. Nuclei were segmented using two-class Otsu thresholding, and cytoplasm was defined as 10 pixels’ extension from nuclei. ERK activity was calculated as the ratio of median mClover fluorescence in cytoplasm to nucleus for each cell. For each biological replicate, ERK activity in 300–600 cells was measured, and each experiment was repeated two times. The removal of imaging artifacts and data processing was performed using custom Python scripts described in [53].

### 2.7. Preparation of Libraries and RNA Sequencing

RNA libraries were generated and sequenced according to [54]. RNA was extracted using a RecoverAll™ Total Nucleic Acid Isolation Kit (Invitrogen). RNA concentrations were measured with a Qubit RNA Assay Kit, and an Agilent 2100 bioanalyzer was used to measure the RNA Integrity Number (RIN). The depletion of ribosomal RNA was performed using an RNA Hyper Kit (Roche), and then library concentrations and fragment length distributions were measured with Qubit (Life Technologies) and Agilent Tapestation (Agilent), respectively. The RNA sequencing was performed using an Illumina NextSeq 550 engine for 50 bp single-end reads and 27–39 million raw reads per sample using standard protocol.

### 2.8. RNA Sequencing Data Processing

Gene expression profiles were initially processed according to [54]. Differential expression analysis was performed using DESeq2 [55]. Genes that were considered differentially expressed had to pass a threshold of Benjamini–Hochberg false discovery rate (FDR)-adjusted *p*-values < 0.05. Volcano plots were visualized using R package EnhancedVolcano (v. 1.16.0).

GO enrichment analysis was conducted using R packages clusterProfile (v.4.2.1) and org.Hs.eg.db (v.3.8.2). We used FDR-adjusted *p*-values < 0.05 as a cutoff value for filtering pathways and GO terms. PCA and visualization were performed for log10-transformed counts of all genes using pca2d R (v.3.6.2) and prcomp software. Gene-regulatory network analysis was performed as described in [56]. Pathway activation levels (PALs) were calculated and visualized with the Oncobox bioinformatical platform [57]. The molecular function of pathway components was algorithmically annotated according to [58]. Testing of the intersection significance was performed according to [59]. EGFR and MAPK signaling pathways were visualized using Oncobox software [60].

### 2.9. Statistical Analysis

Statistical analysis was performed using the GraphPad PRISM 6.0 software (GraphPad Software Inc.), and values of *p* < 0.05 were considered statistically significant. The data were represented as the mean ± SD of at least three experiments performed on different days.

To calculate IC50, the median-effect dose that inhibits the system under study by 50%, we used the equation: IC50 = D/(100/y − 1)^1/m^, where D is the concentration of a drug; y is cell viability (%); and m is the coefficient signifying the shape of the dose–effect relationship.

The significance of pathway activation levels (PALs) was established using the Oncobox pathway analysis method [60] for 1611 molecular pathways containing 10 or more gene products extracted from the public databases [58] using the original software [57]. For PAL calculations, each sample expression profile was normalized on mean geometrical levels of the gene expression for all samples in the dataset under investigation.

We used Benjamini–Hochberg FDR-adjusted *p*-values < 0.05 as a cutoff value for filtering and selecting differentially expressed genes, pathways and GO terms.

To test whether a given number of common differential genes or pathways between the two of three intersecting datasets is significant, 1000 random intersections were performed according to [59]. In every case, two/three random samples from two/three corresponding gene sets of the respective datasets were taken. Then, these random samples were intersected for each iteration, and 1000 numbers of random common genes were obtained. The *p*-value of intersection significance was calculated as a fraction of random numbers equal or higher than the experimentally observed number of common genes.

## 3. Results

To investigate human blood serum crosstalk with EGFR targeted drugs and to assess donor-to-donor variation in such effects, we measured the A431 cell growth rate in the presence of human serum taken from 23 human donors. In order to study underlying molecular mechanisms, we then assessed the cell cycle distribution, activity of ERK1/2 kinases, and profiled gene expression of A431 cells under different drug, EGF, and serum treatment conditions.

### 3.1. Human Blood Serum Donor Specifically Affects A431 Cell Growth Rate

Tumor response to therapeutics can be affected by various molecular factors present in the patient’s body. They can affect the anti-proliferative action of the EGFR-targeted drugs both in vivo and in vitro. In this study, we measured the influence of individual human peripheral blood serum samples on the cell growth rate of epidermoid squamous carcinoma A431 cells in the presence of EGFR-targeted drugs.

First, we studied the influence of human blood serum, in the absence of drugs, on the cell line A431 growth rate (Figure 1). This cell line was selected because it is a classical model for studying effects of EGFR-targeted drugs, including erlotinib and cetuximab, on the EGFR-overexpressing cancer cells [61,62,63]. The A431 cell line is known to express, on average, more EGFR molecules than other cell lines (e.g., ~3 × 10^6^ EGFR molecules per cell versus ~2 × 10^4^ and ~1 × 10^5^ in HEK273 and COS-7 cells, respectively [64]), and it is highly sensitive to EGFR-targeted drugs [65,66].

We used human peripheral blood serum specimens obtained from 23 healthy female human donors. For all the samples tested, the growth media contained heat-inactivated fetal bovine serum (FBS). We, therefore, assessed cell growth rate in the medium containing 5% heat-inactivated FBS and supplemented with 5% human serum for individual donors (or with 5% of additional heat-inactivated FBS for the controls). To implement this study design, we first investigated how A431 cell growth depends on the FBS concentration (1 to 33%) in the growth medium. We found that cell growth rate does not significantly depend on the FBS concentration in a range between 5 and 25% (Appendix A). Thus, cells grown in 5% and 10% FBS could be compared in the same experiment.

We found that in the presence of 5% human blood serum, the A431 growth rate varied by donor with the average of ~60% compared to the FBS-only controls. Noteworthy, for the serum samples from two donors (sd21 and sd4), we detected ~7-fold and 20-fold decreased cell growth rates, respectively (Figure 1). These results are generally in line with our previous findings where considerable effects on A431 cell growth were detected for 2.5% blood serum samples obtained from seven independent human donors [42].

### 3.2. Human Blood Serum Samples Can Antagonize Effects of EGFR-Targeted Drugs on A431 Cell Growth

We then measured A431 cells’ growth rate in the presence of human donor blood sera and EGFR-specific targeted drugs: low molecular mass tyrosine kinase inhibitor erlotinib and therapeutic antibody cetuximab. Growth rates were normalized to the absence of drugs with only FBS present in the culture medium (no-drug level).

Physiological erlotinib concentration under standard therapeutic regimens is about 500 nM [67,68]. When only FBS was present, the erlotinib concentration required for 50% A 431 growth rate inhibition (IC50) was ~330 nM (Appendix A). We then measured the effects of 5% human serum supplements on 1000 nM erlotinib inhibition of A431 cell growth. This concentration of erlotinib was selected because it almost completely inhibited A431 cell growth (Figure 2a). However, we found that most of the tested human serum samples could statistically significantly rescue cells from growth inhibition by erlotinib (Figure 2a). With human serum added, the average growth rate was ~46% of no-drug level, whereas this proportion was negligible in the presence of FBS only. To determine erlotinib concentration sufficient to inhibit A431 growth in the presence of human serum, we treated media supplemented with sera from four representative human donors with increasing concentrations of erlotinib (Figure 2b). Again, we found that human serum could strongly interfere with cell growth inhibition by erlotinib, which resulted in a 2–5-fold increased IC50 (Figure 2b). The combination index [69] for erlotinib and human serum varied from 1.5 to 6.7 in 23 tested blood samples, which corresponds to moderate and strong antagonism. This is in line with our previous findings, where we showed that at erlotinib concentration of as high as 1200 nM (~2.5 fold higher than normal therapeutic concentration), the 2.5% mixed human serum sample for seven donors increased A431 clonogenity by approximately six times (~4% versus ~23%) [42].

Similarly, we then investigated the effects of cetuximab on A431 cell growth rate in the presence of human blood serum (Figure 3a). In the absence of human serum, cetuximab IC50 was ~0.18 µg/mL (Appendix A), and at a concentration of 1 µg/mL, cetuximab completely arrested A431 growth in FBS-only medium. In contrast, most of the human serum samples antagonized cetuximab’s inhibitory effects and statistically significantly increased the cell growth rate. Under these conditions, all 23 human serum samples tested showed a capacity to diminish the cetuximab inhibition of the A431 growth rate (Figure 3a). This has also dramatically shifted the IC50 values of cetuximab for the human serum samples. Furthermore, even at the highest cetuximab concentration tested (6 µg/mL, ~50 times the IC50) in the presence of human serum, a significant fraction of cells (~10%) persisted and was capable of proliferating (Figure 3b). Overall, human serum could essentially decrease A431 growth inhibition by cetuximab and augment cetuximab IC50 by ~5–20-fold.

The combination index for cetuximab and 5% human serum varied between 3.2 and 15 among 23 human samples tested with an average value of 10.2, which corresponds to strong antagonism [69]. This finding is in good agreement with our previous observation that at 1 µg/mL cetuximab, the colony formation rate varied between 40% and 100% in the presence of seven human serum samples, while only a few colonies could be detected when no human serum was added [42]. Here, the combination index [69] for cetuximab and human serum varied between 3.2 and 15 among 23 experimental human serum samples tested with average value of 10.2, which corresponds to strong antagonism according to our estimation of cell growth rate (Figure 3a). Taken together, our results demonstrate that human blood serum can abolish the cell growth arrest caused by EGFR-targeted drugs: both monoclonal antibodies and low molecular mass inhibitors.

### 3.3. Dual Effects of EGF on A431 Cell Growth and Resistance to EGFR Inhibitors

It can be hypothesized that at least partly, the cell growth-promoting effects of human serum may be related to EGF, which is a major EGFR ligand from blood flow. Indeed, it was found previously for several HER-positive cell lines that EGF and another HER ligand NRG1 can interfere with the activities of lapatinib (drug targeting both EGFR and HER2) and cetuximab [39,40,41,42]. Consequently, here, we explored the effects of recombinant human EGF on the cell growth rate of A431 cells in the presence of EGFR-targeted drugs.

EGF concentrations in human blood can vary widely among individuals, with the reference range of ~0.3–1.7 ng/mL, while the concentration of other HER ligands is much lower: TGF-alpha ~0.01 ng/mL; epiregulin ~0.3 ng/mL; betacellulin ~0.2 ng/mL [42,70]. EGF is known to have a dual activity on cell growth by stimulating proliferation at low, ~0.1 ng/mL, concentrations, or by inhibiting proliferation at greater concentrations of 70 ng/mL and higher in human MDA MB-468 and A431 cells [23,71]. Recently, we reported on the counterintuitive EGF cell growth inhibition of A431 cells under EGF concentration in the above physiological concentration range [42]. We showed that EGF abrogates the growth of squamous carcinoma A431 cells at concentrations exceeding 0.8 ng/mL with IC50 at ~0.4 ng/mL (Appendix A).

In agreement with the previous reports [23,68], we found that the combined treatment of A431 cells by the EGFR inhibitors and EGF restored their growth (Figure 4). This phenomenon could be due to the interplay of the EGFR inhibition and activation mechanisms [72]. The results of dose-dependent cell growth rate changes of A431 cells at the 0–36 µg/mL concentration range of cetuximab and fixed EGF concentrations (0, 5, and 10 ng/mL) are shown in Figure 4b. The physiological blood concentration of cetuximab during head and neck squamous cell carcinoma treatment is ~30 µg/mL [73]. In our experiments, the cetuximab IC50 was ~0.18 µg/mL. However, even at cetuximab concentrations of up to 6 µg/mL (which more than 30-fold exceeds IC50), the no-drug level of cell growth was almost restored in the presence of 0.5–3 ng/mL EGF (Figure 4b). Furthermore, cell growth was not arrested even at a cetuximab concentration of 36 µg/mL and was ~10, 59, and 63% of the no-drug FBS control level for EGF concentrations of 0.5, 1.5, and 3.1 ng/mL, respectively.

Similarly, we found that EGF can strongly rescue A431 cells from growth inhibition by erlotinib. Even at low concentration of 0.15 and 0.3 ng/mL, when EGF moderately inhibits cell growth, its presence has significantly increased the IC50 value of erlotinib (Figure 4a). At a higher concentration of 5 ng/mL, when EGF itself strongly suppresses A431 growth, in the presence of erlotinib, it restores cell growth to rates comparable with the no-drug level.

As for the bell-shaped curves, an accurate assessment of IC50 can be ambiguous, so we also for the first time assessed a combination index [69] for the crosstalk of EGF and erlotinib and for EGF and cetuximab. The observed combination index values of ~17 and ~23 correspond to the strong antagonism of EGF with erlotinib and with cetuximab, respectively. Thus, our data confirmed that both EGFR inhibitors tested and EGFR ligands can inhibit the growth of A431 squamous carcinoma cells, whereas together, they can counterbalance each other and release cell proliferation.

We measured the EGF concentration in human sera used here by ELISA; it varies between 0.7 and 1.4 ng/mL. Thus, in our experiments, the EGF concentration in the growth media supplemented with 5% human serum does not exceed 0.07 ng/mL. We conclude that the human blood serum influence on the erlotinib and cetuximab impact (Figure 2 and Figure 3) cannot be explained by endogenous EGF action.

### 3.4. In A431 Cells, Cetuximab and Erlotinib Cause Alterations in Passing through the Cell Cycle which Can Be Abolished by the Presence of EGF or Human Blood Serum Samples

EGFR-targeted drugs are known to inhibit cell cycle progression by arresting the G0/G1 phase transition [45]. In this study, we measured the cell cycle distribution in A431 cells by fluorescence-activated cell sorting (FACS) analysis. We found that EGF treatment at a concentration of 8 ng/mL has a statistically significantly increased proportion of cells in the G0/G1 phase compared to the no-drug level (68.7% versus 61.6%, plus 7.1%); see Figure 5. This finding is in line with previous studies [72]. In 700 nM erlotinib-treated cells, we observed a greater G0/G1 shift of plus 19.8% of cells compared with the no-drug control level (Figure 5a, Appendix A, Appendix A). Note that such treatment also inhibited cell growth to only 7% from the no-drug level. When EGF was added along with erlotinib, the growth rate restored to ~28% of the no-drug level. At the same time, the proportion of G0/G1 cells did not significantly shift back to the no-drug level, and it remained increased by ~20% for both 2 and 8 ng/mL EGF concentrations, which was similar to the absence of EGF.

We then measured the cell cycle distribution of A431 cells in the media supplemented with the sera of four human donors (Figure 5b). For all four samples tested, the proportion of G0/G1 cells has not significantly changed, whereas the proportion of cells in the S phase has slightly increased by ~1.3% (Appendix A).

To our knowledge, the influence of human serum samples on cell growth inhibition by erlotinib or cetuximab at the level of cell cycle analysis has not been reported previously. In the presence of human serum (mean values for four serum samples) and erlotinib, the A431 G0/G1 phase shifts from the no-drug level was ~10% instead of ~20% with erlotinib only (Figure 5). At the same conditions, cell growth rates were ~50% and ~2%, respectively (Figure 2).

In the case of 0.6 µg/mL cetuximab, we observed a plus ~15% G0/G1 shift from the no-drug level (Figure 5c, Appendix A). This treatment inhibited cell growth to only ~8% of the no-drug level. In turn, when 8 ng/mL EGF was added along with cetuximab, the cell growth rate increased to 82% of the no-drug level, and this was accompanied by a G0/G1 shift back to the no-drug level. This effect of EGF seems to be dose-dependent, as 2 ng/mL EGF could not completely shift G0/G1 value back to the no-drug control level (Figure 5c). This effect can be related to the previously found EGF-mediated upregulation of p21(Cip1) and cetuximab-mediated upregulation of p27(Kip1) [72].

In the presence of human serum (mean values for four donors) and cetuximab, the growth rate of A431 cells was ~80% of no-drug level instead of 8% with cetuximab only. This is in good agreement with our finding that cetuximab + human serum causes only a 6% G0/G1 phase shift, while cetuximab-only treatment results in a 15% shift from the no-drug level (Figure 5d).

Taken together with the above findings for erlotinib and cetuximab, these results can provide mechanistic clues toward the understanding of interference of EGFR-targeted drugs with human blood serum, which can be somewhat similar to the activities of EGF.

### 3.5. Cetuximab and Erlotinib Treatment Inhibits ERK1/2 Activity in A431 Cells: This Effect Is Abrogated by Human Blood Serum or EGF

ERK proteins are important downstream effectors on the crossroads of multiple signaling pathways [74], and their activities were shown to be indicative for the whole EGFR pathway activation [75]. Thus, we evaluated whether EGFR inhibition by targeted drugs in A431 cells can be connected with the differential activation of ERKs. To this end, we created transformed A431 cells with the stable expression of kinase translocation reporter (KTR) of ERK1/2 activity (ERK-KTR). This reporter is based on the fluorescent protein that is transported into the cell cytoplasm after its phosphorylation by ERKs. This allows a vital measurement of ERK activity in single cells by calculating the cytoplasm to nucleus ratio of the fluorescent signal [51], as shown in Figure 6a. Previously, we showed that this reporter can be used to evaluate the ERK-dependent action of cancer drugs and growth factors [53,76].

We employed two ERK-specific inhibitors, ulixertinib and SCH772984, to control if A431 growth is associated with ERK1/2 activity. Ulixertinib (BVD-523) [77] and SCH772984 [78,79] are selective low molecular mass inhibitors that compete with ATP for binding by ERK1/2 proteins. In vitro ulixertinib treatment was shown to result in reduced proliferation and enhanced caspase activity in cultured cells [80].

In this study, we measured cell growth rates in the presence of both ERK inhibitors and found that they can essentially decrease the A431 growth rate with IC50 for ulixertinib ~550 nM and ~150 nM for SCH772984. In the ERK-KTR reporter A431cells, the application of both inhibitors resulted in drastically decreased levels of ERK activation compared to the control level (Figure 6b). This confirmed the adequacy of the ERK-KTR reporter system for functional tests in A431 cells and justified its further use in our molecular studies of EGFR regulatory interplay.

We found that A431 treatment with cetuximab has dramatically decreased ERK activity. When cells were treated with 0.5 to 2.6 µg/mL of cetuximab for 24 h, the ERK activity dropped to less than 7% of its no-drug level. We, therefore, concluded that cetuximab inhibits ERK activity to almost the same extent as specific ERK inhibitors ulixertinib and SCH772984 (0.83 and 0.85, respectively; see Figure 6c,d). At lower concentrations (0.16 and 0.5 µg/mL), cetuximab inhibited both the ERK activity and cell growth rate in a dose-dependent manner (Figure 3b). Thus, we conclude that the ERK activity correlates with cell growth in the presence of cetuximab (Figure 6f).

In agreement with that, 200–1000 nM erlotinib treatment also dose-dependently inhibited ERK activity. At high erlotinib concentrations, ERK activity also dropped to the level previously registered for the above ERK-specific inhibitors (Figure 6e), which coincided with the dose-dependent decrease in cell growth rate (Figure 2b). Thus, we conclude that ERK activity correlates with cell growth in the presence of both EGFR-targeted drugs (Figure 6f).

When added alone, EGF also decreased the ERK activity when applied for 24 h at 5 and 10 ng/mL (Figure 6c–e). EGF reduces ERK activity to 36%, and it remains significantly higher than the level found for the two ERK inhibitors tested. Although both cetuximab and EGF inhibited ERK activity when added alone, when added together, they failed to inhibit ERK and restored its activity to the no-drug control level (Figure 6c,d). This is consistent with the observed reduced toxicity of cetuximab in the presence of EGF, and it suggests an ERK-dependent mechanism of EGF–drug interference influencing A431 cell growth.

In further ERK activity tests, we used the mixed human serum samples obtained by pooling materials of eight donors: sd 2, 14, 15, 17, 18, 19, 20, and 21. We found that the human blood serum alone did not essentially influence ERK activity, which remained at the level of ~85% of the no-drug level (Figure 6c–e). When the cells were treated by both 5% human serum and increasing concentrations of cetuximab (0.4–3 µg/mL), the ERK activity was measured as high as 62–71% (Figure 6c,d). Under the same experimental conditions, the A431 growth rate was ~30–82% in the presence human serum and 0–16% without human serum, depending on the cetuximab concentration (Figure 3a).

Thus, again, we observed a match between the cell growth and ERK activity. Similar results were obtained for combined human blood serum + erlotinib treatment (Figure 6e): erlotinib only abolished cell growth and inhibited ERK activity. Under the same conditions, ERK activity was between 53 and 85% of the no-drug level in the presence of 300–900 nM erlotinib and human serum. In the absence of human serum, only 0–43% cell growth was detected, depending on erlotinib concentration (Figure 2b).

Taken together, these findings most probably suggest an ERK-dependent mechanism of human blood serum interference with cetuximab and/or erlotinib, which has a considerable impact on A431 cell growth and survival.

### 3.6. Transcriptome Changes Connected with the Interplay of EGF, EGFR Inhibitors, and Human Serum in A431 Cells

To assess molecular mechanisms underlying human serum and/or the EGF rescue of A431 cells treated with EGFR-targeted drugs, we performed transcriptome analysis by using RNA sequencing of treated cell culture samples. As in the ERK functional assay, we used mixed human serum samples obtained by pooling the materials of eight donors. For every functional condition, sequencing experiments were performed in triplicate. Fold change values of differentially expressed genes (DEGs) to the control no-drug level (FDR-adjusted *p*-value < 0.05; fold change > 2) are given in Appendix A; complete sequencing data are deposited at ID 966,187—BioProject—NCBI (nih.gov).

We found that the human blood serum treatment of A431 cells resulted in an upregulation of 20 and downregulation of five genes. Interestingly, 13 (65%) of these upregulated genes were in turn downregulated by both cetuximab and erlotinib treatments.

We found that 48 h treatment with 0.6 µg/mL cetuximab, which completely arrests A431 cell growth, causes a downregulation of 932 and upregulation of 339 genes (Figure 7a). Thus, cetuximab treatment essentially changes the transcriptomic profile, and we then checked if cetuximab inhibits A431 cell growth reversibly or whether treated cells undergo death or senescence. We ensured that cell growth inhibition by cetuximab for three days is reversible (Appendix A).

We observed that EGF added in combination with cetuximab shaped transcriptomic profiles very close to the no-drug level with only seven DEGs identified (Figure 7a). This reflected the cell growth rate, which was also similar (~86%) to the no-drug control; note that cetuximab concentration was 5-fold larger than the IC50 value, and EGF concentration was 10-fold larger than the IC50 value.

A431 cells treatment with cetuximab completely arrests cell growth, whereas the addition of a 5% human blood serum sample could restore the A431 growth rate to 96% of the no-drug level (Figure 8a). In line with this observation, we found that 75% of DEGs, whose expression was differential in the cetuximab-only treatment condition, became non-differential in the presence of cetuximab plus human blood serum (253 upregulated and 685 downregulated DEGs, Figure 8b,c). Thus, we concluded that the differential expression of those DEGs is most likely sufficient to abrogate or restore A431 cell growth. For brevity, hereinafter, this group of genes is called the Cetuximab core gene group.

The treatment of A431 cells with 800 nM erlotinib completely arrested A431 cell growth (Figure 8a) and resulted in 1566 DEGs (665 upregulated genes and 901 downregulated; see Figure 7b). We ensured that cell growth inhibition by erlotinib is reversible (Appendix A). The addition of EGF in combination with erlotinib has removed 39% of those DEGs (313 upregulated and 300 downregulated genes, Figure 8d,e). Physiologically, this corresponds to a ~58% recovery of the no-drug level of cell growth.

When a mixed human blood serum sample was present in combination with erlotinib, A431 cell growth was restored to 67% of the no-drug level (Figure 8a). In comparison to erlotinib-specific DEGs, in the presence of human serum plus erlotinib, 83% of all DEGs became non-differential (520 upregulated and 777 downregulated genes, Figure 8d,e). It is sufficient to restore A431 cell growth. We hypothesize those genes which were differentially expressed in the presence of erlotinib only but became not differential in the presence of erlotinib plus human serum are most strongly associated with cell growth arrest by erlotinib. Hereinafter, we called this group of genes the Erlotinib core gene group.

### 3.7. Molecular Pathways Associated with Molecular Interplay of EGF, EGFR Inhibitors, and Human Serum

We used Gene Ontology (GO) analysis to identify functional terms enriched in the Cetuximab core gene group (Figure 9). We found that the most significantly enriched terms were associated with the activity of ribosomes, metabolism of retinoids, cell–substrate adhesion, regulation of kinase activity and wound healing, and regulation of viral process. The same top 20 GO terms were also found for the cetuximab-only group of DEGs.

For the Erlotinib core gene group, the most significantly enriched GO terms were partly overlapping with the Cetuximab core gene group and were connected with the activity of ribosomes, cell–substrate adhesion, nephron epithelium development, and NF-kappaB-inducing kinase activity (Figure 9). For erlotinib-only treatment, the same GO terms were identified. Hence, these major GO terms are highly likely to be associated with cell growth arrest by the EGFR-targeted drugs.

We then calculated the pathway activation levels (PALs) of 3044 human molecular pathways and visualized the results using the OncoboxPD online tool [57]. The top-ten most strongly activated and inhibited pathways for Cetuximab and for the Erlotinib core gene groups are shown in Figure 10a,b, respectively. Interestingly, exactly the same pathways were identified in Cetuximab vs. no and Erlotinib vs. no groups. Thus, human blood serum essentially abrogates the differential activities of these pathways, which is in agreement with the growth rate measurements (Figure 8a).

Among the top ten most strongly activated pathways in the Cetuximab core gene group, four also coincided in the Erlotinib core gene group (Figure 10), namely, “NCI urokinase type plasminogen activator uPA and uPAR-mediated signaling pathway (cell adhesion)”; “NCI validated transcriptional targets of AP1 family members Fra1 and Fra2 main pathway”; “Reactome telomere extension by telomerase main pathway”; “Reactome the nlrp1 inflammasome main pathway”. Two other pathways belonged to the top 20 most strongly activated pathways in the Erlotinib core gene group (Appendix A); finally, four remaining pathways were also strongly activated in the Erlotinib core group. This may reflect inhibition by cetuximab and erlotinib of the same molecular target, EGFR.

In turn, among the top 10 most strongly activated pathways in the Erlotinib core gene group, four were among the top 10 most activated pathways in the Cetuximab core group as mentioned above, five were among top 20 most activated pathways in the Cetuximab group, whereas one remaining pathway, “NCI FOXA2 and FOXA3 transcription factor networks main pathway”, was not statistically significantly activated in the Cetuximab core gene group.

Among the top ten most strongly inhibited pathways in the Cetuximab core gene group, four pathways coincided with the Erlotinib group (Figure 10), namely, “deoxy-alpha-D-ribose 1-phosphate degradation”; “biocarta sprouty regulation of tyrosine kinase signals pathway (cell migration)”; “NCI S1P5 pathway (negative regulation of cAMP metabolic process)”; “Reactome activation of BMF (Bcl2-modifying factor) and translocation to mitochondria main pathway” and six other pathways belonged to the top 20 most strongly activated pathways in the Erlotinib group. In turn, among the top- ten most strongly inhibited pathways from the Erlotinib group, four pathways coincided with the Cetuximab group (see above), and six other pathways were statistically significantly inhibited also in the Cetuximab group.

We then compared the pathway activation charts of the EGFR signaling pathway calculated for the A431 cells treated with cetuximab in comparison to no-drug condition (Figure 11a) or for cetuximab core genes (Figure 11b). Similarly, such a comparison was performed for erlotinib in comparison to the no-drug condition and for erlotinib core genes (Figure 11c,d). We found that the activation levels of the EGFR pathway components significantly differ between cells treated with cetuximab and in the no-drug condition (Figure 11a). The major components of the EGFR pathway, MEK, STATs, Ras, PI3K, and Akt, were downregulated by cetuximab. All these nodes were also downregulated by erlotinib treatment (Figure 11c). Of note, all these nodes and most of the other EGFR pathway genes were included in cetuximab (Figure 11b) and erlotinib (Figure 11d) core gene sets. This suggests a strong association of the EGFR pathway with the cell growth inhibitory mechanisms by both drugs. Another most strongly associated pathway for both drugs was the MAPK signaling pathway, where core gene sets accounted for ~100% of the nodes (Figure 12b,d).

Interestingly, both human blood serum and EGF decreased the overall activities of the EGFR pathway in treated A431 cells: from the neutral pathway activation level (PAL) to negative values of −1.7 and −2.9, respectively, where the difference with the control for the serum was not statistically significant (Figure 11e,h). At the same time, treatment with cetuximab and erlotinib resulted in a much stronger downregulation of the EGFR pathway, as reflected by PAL values of −14.5 and −12.1, respectively (Figure 11a,c). On the other hand, combined treatment with drugs plus human serum has raised these EGFR pathway PAL values to −4.4 and −6.2, respectively (Figure 11f,g). At the same time, the combined treatment of drugs plus EGF in comparison to drugs plus serum resulted in even stronger effects for cetuximab (PAL −1.1; Figure 11i) but virtually no effect for erlotinib (PAL −14.7; Figure 11j).

The same activation trend is reproduced also for the MAPK signaling pathway (Figure 12). Human serum and EGF produce little changes in the pathway regulation (PAL 1.3 and 2.3, respectively); see Figure 12e,h. The addition of drugs results in a sharp decrease in MAPK pathway activation, as reflected by PAL −11.2 and −12.6 for cetuximab and erlotinib, respectively (Figure 12a,c). At the same time, this decrease is strongly attenuated when drug is added in combination with human serum, when the PAL is −1.8 and −4.6, respectively (Figure 12f,g). When drug is added in combination with EGF, the drug rescue effect for cetuximab becomes even stronger with PAL 3.9 (Figure 12i), whereas it remains negligible for erlotinib (Figure 12j).

Overall, our findings at least partly explain the mechanisms of A431 cells rescue from EGFR-targeted drugs. When mediated by human serum, the growth-suppression effect caused by cetuximab or erlotinib is almost completely abolished. This correlates with the reactivation of the EGFR and MAPK pathways (Figure 11 and Figure 12), with restored ERK1/2 activities in vivo (Figure 6), and with only a moderately changed proportion of A431 cells in the G1 phase of the cell cycle compared to no-drug conditions (Figure 5).

In contrast, when mediated by EGF, this also works in the same way for the cetuximab treatment, but it acts differently for erlotinib. In combination with EGF, erlotinib showed smaller drug rescue effects compared to cetuximab (Figure 4). In EGF plus erlotinib-treated cells, we observed a bigger proportion of cells in the G1 phase (Figure 5) and smaller ERK1/2 activation (Figure 6). We could also detect no reactivation of the EGFR and MAPK pathways (Figure 11 and Figure 12).

Treatment of A431 cells with cetuximab or erlotinib results in strongly overlapping yet not identical sets of DEGs (Figure 8). Interestingly, most of these genes returned to normal expression levels when mixed human serum sample was added to the medium (Figure 9). In the case of EGF added to medium with cetuximab, all cetuximab-specific DEGs became non-differential (Figure 9), and the total number of DEGs amounted to only seven. This was not the case for the co-treatment of A431 cells with EGF + erlotinib, which resulted in a smaller yet still high number of DEGs (1177), which was only ~30% less than for the erlotinib-only treatment (Figure 8). At the same time, the number of DEGs for erlotinib + human serum was only 299, which is approximately four times lower. Taken together, these findings clearly indicate strongly overlapping mechanisms for human serum and EGF in rescuing A431 cells from cetuximab and only partly overlapping but however different mechanisms of human serum and EGF when supplementing erlotinib. This most probably reflects different mechanisms of action of these drugs, where the external part of EGFR is the only known binding partner for monoclonal antibody cetuximab, and its kinase domain is targeted by the small molecular mass inhibitor erlotinib. However, as stated in the FDA documents, “The mechanism of clinical antitumor action of erlotinib is not fully characterized. Erlotinib inhibits the intracellular phosphorylation of tyrosine kinase associated with the epidermal growth factor receptor (EGFR). Specificity of inhibition with regard to other tyrosine kinase receptors has not been fully characterized” (https://www.accessdata.fda.gov/drugsatfda_docs/label/2008/021743s010lbl.pdf (accessed on 15 July 2023)).

Thus, the cetuximab and erlotinib mechanisms of action may be not identical, with more complex molecular and regulatory networks affected by erlotinib. We speculate that the above-mentioned features may be responsible for the smaller cell growth rescue effects of EGF for erlotinib in comparison to cetuximab.

## 4. Discussion

The prediction of individual patient response to drug therapy is a challenging task in modern oncology. Most frequently, it involves genetic, epigenetic, or transcriptomic markers of individual tumors [34,35,81,82,83]. However, not only “intrinsic” tumor molecular properties can be relevant. Various effects of blood serum components were assessed in several cancer clinical studies [30,84]. Nonetheless, the influence of human serum on the specific effects of targeted drugs has not been yet sufficiently investigated. The elucidation of underlying molecular mechanisms requires research involving cell culture models [39,40,41] and in-depth molecular pathway analysis [60,85,86]. Recently, we found that human peripheral blood can strongly modulate repression by HER-specific drugs of the colony formation ability by human cancer cells [42,44].

In this study, using human squamous carcinoma cell line A431, we showed a dramatic and donor-specific influence of human blood serum on the inhibition of cell growth rate by EGFR-targeted drugs. Our clonogenicity and cell count assays demonstrated that human serum and EGFR-targeted drugs have reciprocal antagonistic effects. Thus, in our experiments, the effective concentrations of EGFR-targeted drugs could be affected 5–25-fold by the addition of different human serum samples. This observation may be considered when experimental drugs are being preclinically tested for their antitumor activity. In addition, this remarkable interference of human sera with drugs could be most probably taken into consideration when estimating the potential drug response of an individual patient, as least for the EGFR-targeted medicines.

We found that the majority of tested human serum samples to some extent diminished the effect of EGFR-targeted drugs on A431 cell growth. In particular, 22/23 samples tested significantly increased the cell growth rate in the presence of erlotinib (Figure 2), and 16/23 samples increased the cell growth rate when combined with cetuximab (Figure 3). Such an antagonism with drugs was donor specific. Human blood serum interference with the drug inhibition of cell growth varied significantly from donor to donor for both drugs, whereas the donor-to-donor variation of the combinational index for drugs and human serum was up to about 4.5 times for cetuximab and 4.7 times for erlotinib.

In addition, we showed that the proliferation of A431 cells inhibited by EGFR-targeted drugs could be restored by relatively low physiological concentrations of EGF. For example, in the presence of a sufficient amount of EGF, even high concentrations of cetuximab could not fully inhibit cell growth. An assumption of equilibrium between EGF and cetuximab action cannot explain up to a 100× drug concentration range where cells proliferate in the presence of the fixed EGF concentration (Figure 4b). In turn, for EGF/erlotinib interplay, a gradual increase in erlotinib concentration leads first to an increase in A431 cell growth, and then to a decrease in the cell growth arrest (Figure 4a). Previously, we observed such a bell-shaped curve for A431 colony number vs. EGF concentration at a fixed cetuximab concentration [42].

We also showed that both cetuximab and erlotinib inhibit cell cycle progression by causing the G0/G1 phase arrest (Figure 5), which is in line with the previous studies [45,87,88]. It was reported previously that the inhibition of A431 cells by EGF is associated with an upregulation of p21(Cip1) and G0/G1 phase arrest, but when EGF was added along with cetuximab, then the p21 level was decreasing [72]. In this study, when cetuximab was added along with EGF, the growth rate was restored, and the proportion of cells in the G0/G1 fraction was shifted back toward the no-drug level. However, when erlotinib with EGF was added, the proportion of cells in the G0/G1 fraction remained high and did not return to the no-drug level. In turn, human serum samples could abrogate both cetuximab- and erlotinib-induced G0/G1 phase bias.

The major EGFR downstream target, the Ras/RAF/MEK/ERK (MAPK) signaling pathway, plays a crucial role in the survival, growth and proliferation of cancer cells, and its hyperactivation is responsible for over 40% of human cancer cases [46,89]. Here, we measured the in vivo ERK1/2 activity of drug-, EGF-, and human serum-treated cells to investigate the possible ERK dependence of the molecular mechanisms observed (Figure 6). We found that cetuximab, erlotinib, and EGF inhibit ERK1/2 activity when added alone. However, in combination with EGF, cetuximab failed to inhibit ERK1/2, whereas erlotinib did but to a much lesser extent. In contrast, the mixed human blood serum sample alone did not affect ERK1/2 activity, but it could almost completely restore the ERK1/2 activity of A431 cells to a no-drug level for both cetuximab and erlotinib.

We then investigated the mechanism of EGFR-specific drug activity on A431 cells by RNA sequencing. We revealed as much as 1271 differentially expressed genes (DEGs) in the presence of cetuximab. Outstandingly, the expression of neither of these DEGs was altered when both components, cetuximab and EGF, were added simultaneously to the cell growth medium. When human serum was added along with cetuximab, only a quarter of these cetuximab-specific DEGs remained differentially regulated. Thus, we conclude that the expression of the remaining 75% of the DEGs was sufficient to restore A431 cell growth. Erlotinib-only treatment causes an alteration of the expression of 1566 genes. Human serum added along with erlotinib restored the expression of 83% of erlotinib-specific DEGs. In turn, EGF added with erlotinib restored the expression of only 39% of the erlotinib-specific DEGs.

Both erlotinib and cetuximab molecularly target EGFR, and we observed a significant overlap in the DEG sets induced by both these drugs: out of 339 cetuximab upregulated genes, 86% were also upregulated by erlotinib; of 932 downregulated genes, 69% were also downregulated by erlotinib (Appendix A). Thus, we propose that the genes responsible for the cell growth arrest caused by the inhibition of EGFR could be identified as DEGs when treated by both cetuximab and erlotinib. As such, DEGs whose expression is not affected in the presence of drugs plus EGF or human serum could be considered as the major set of A431 drug-sensitivity genes. We found 127 such genes (94 downregulated and 33 upregulated by both, cetuximab and erlotinib; see Appendix A).

Several possible crosslinks of the core survival gene set with the molecular functions can be mentioned. Both drugs downregulated genes *RIMS1* and *RAB15* that encode proteins of the RAS superfamily. Ras proteins regulate multiple cellular processes including cell cycle progression, growth and survival [90] by acting as the upstream activators of the Raf–MEK–ERK axis [91]. EGFR drugs downregulate *CAMK2N1* and upregulate *IL23A,* which are MAPK/ERK pathway modulators [92,93]. *GLUL* is downregulated by the drugs, whereas its enhanced expression in breast cancer is associated with a larger tumor size and higher expression of HER2. In turn, *GLUL* knockdown inhibits the p38 MAPK and ERK1/ERK2 signaling pathways [94]. Differential expression of the above genes is in good agreement with our finding that both cetuximab and erlotinib are linked to decreased ERK1/2 activity, while a simultaneous treatment with human blood serum or EGF restores the cell growth and activity of ERK1/2 (Figure 6). We also found that both drugs strongly affect the NF-κB pathway, and that this effect is reversed by the addition of either human serum or EGF (Figure 11; see node “NFKB”). This pathway is one of the major regulators of cell survival and immune response [95].

In addition, by calculating molecular pathway activation levels, we provide evidence that cell resistance to EGFR-targeted drugs treatment in the presence of human serum can be due to the reactivation of EGFR and MAPK pathways. We also suggest that erlotinib and cetuximab have different yet significantly overlapping molecular mechanisms of their activity on squamous cell carcinoma A431 cells, which is reflected by the different perturbations of the intracellular molecular network and somewhat different cellular physiology responses.

Our results suggest that it could be useful to check drug impacts in preclinical studies in the presence of human blood serum. Finally, we speculate that the in vitro testing of drug activity in the presence of blood serum of individual patients and a model cancer cell line can be beneficial for the personalization of EGFR-specific therapeutics.

## 5. Conclusions

We demonstrate that human serum abolishes cell growth rate inhibition by the EGFR-specific drugs cetuximab and erlotinib. Human blood serum-mediated A431 resistance to EGFR drugs can be largely explained by the reactivation of MAPK signaling cascade. Human serum treatment restores ERK1/2 activity inhibited by cetuximab and erlotinib. Our results demonstrate that drug inhibitory effects estimations could be erroneous when drug activities are measured in growth media containing only bovine serum. An undesirable diminishing of drug impact by human blood sera must be taken into account in preclinical studies.

## Figures and Tables

**Figure 1 cells-12-02022-f001:**
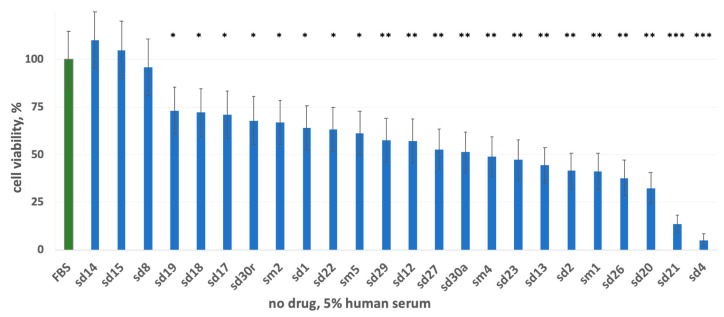
Growth rate of A431 cells in the media containing 5% FBS supplemented with 5% human blood serum samples. Growth rate is normalized to the FBS-only medium, 10% FBS (column “FBS”). Asterisks stand for statistically significant differences between FBS and human serum samples: *, *p* < 0.05; **, *p* < 0.01; ***, *p* < 0.001.

**Figure 2 cells-12-02022-f002:**
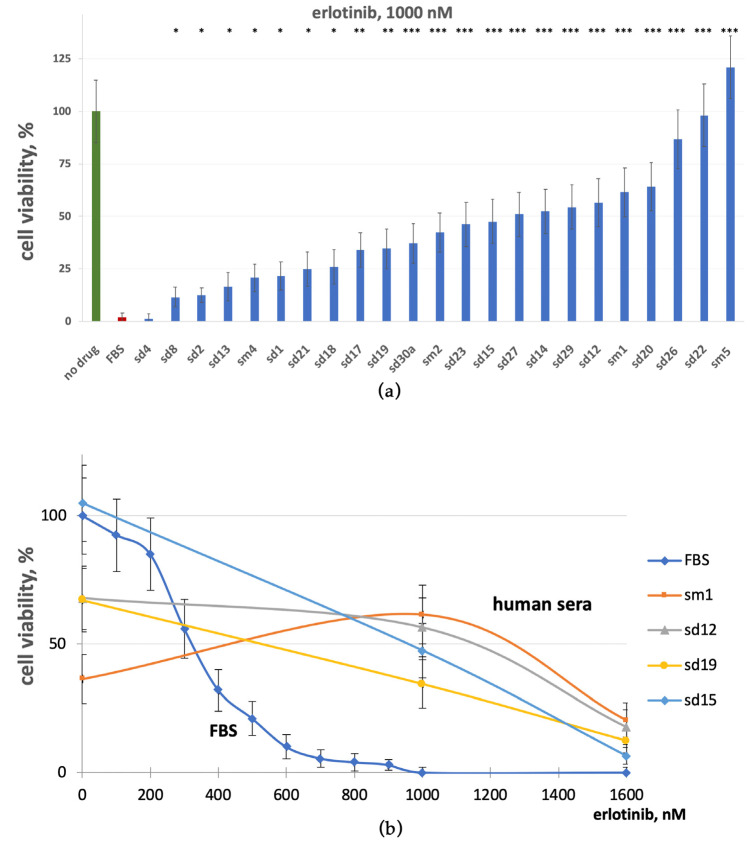
Growth rate of A431 cells in the presence of erlotinib and human serum. (**a**) Growth medium is supplemented with 1000 nM erlotinib and 5% human donor serum samples plus 5% FBS or 10% FBS (FBS). (**b**) Growth medium is supplemented with increasing concentrations of erlotinib (0–1600 nM) and 5% human serum samples (sm1, sd12, sd19, and sd15) plus 5% FBS or 10% FBS (FBS). Bars represent the average cell growth rate for each donor sample calculated from three replicates normalized to no-drug conditions with FBS-only growth media (no drug). Asterisks stand for statistically significant differences between FBS + erlotinib and Human serum + erlotinib samples: *, *p* < 0.05; **, *p* < 0.01; ***, *p* < 0.001. Dose–response curves for each donor sample calculated from three replicates normalized to no-drug conditions with FBS-only growth media.

**Figure 3 cells-12-02022-f003:**
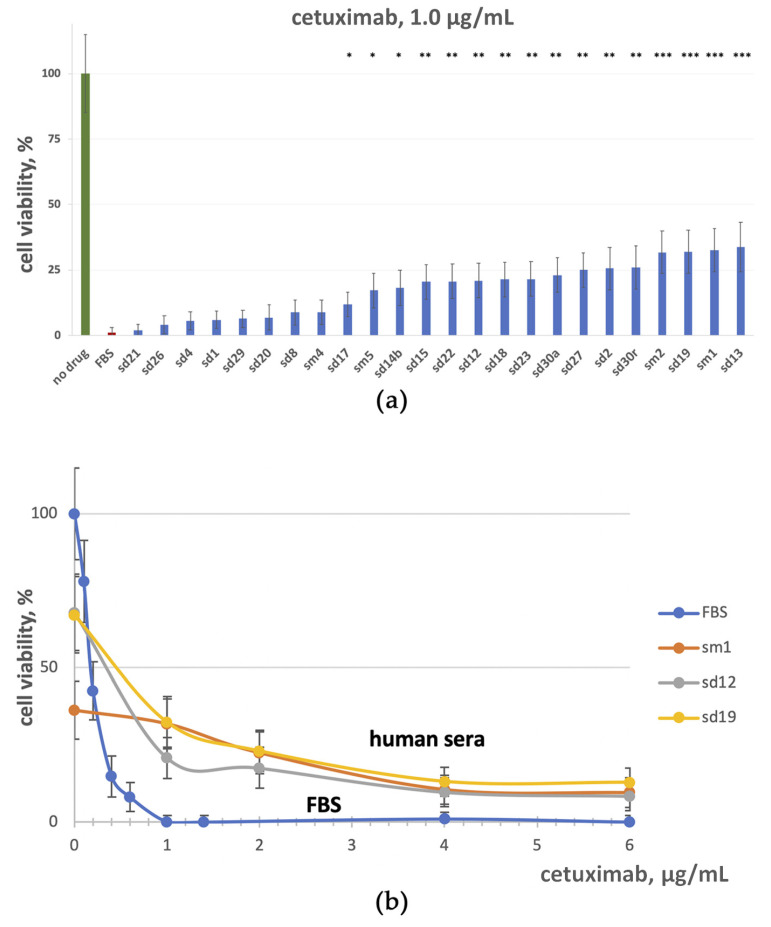
Growth rate of A431 cells in the presence of cetuximab and human serum. (**a**) Growth medium is supplemented with 1 ug/mL cetuximab and 5% human donor serum samples plus 5% FBS or 10% FBS (FBS). (**b**) Growth medium is supplemented with increasing concentrations of cetuximab (0.1–6 µg/mL) and 5% human serum samples (sm1, sd12, and sd19) plus 5% FBS or 10% FBS (FBS). Bars represent the average cell growth rate for each donor sample calculated from three replicates normalized to no-drug conditions with FBS-only growth media (no-drug). Asterisks stand for statistically significant differences between FBS + cetuximab and Human serum + cetuximab samples: *, *p* < 0.05; **, *p* < 0.01; ***, *p* < 0.001. Dose–response curves for each donor sample calculated from three replicates normalized to no-drug conditions with FBS-only growth media.

**Figure 4 cells-12-02022-f004:**
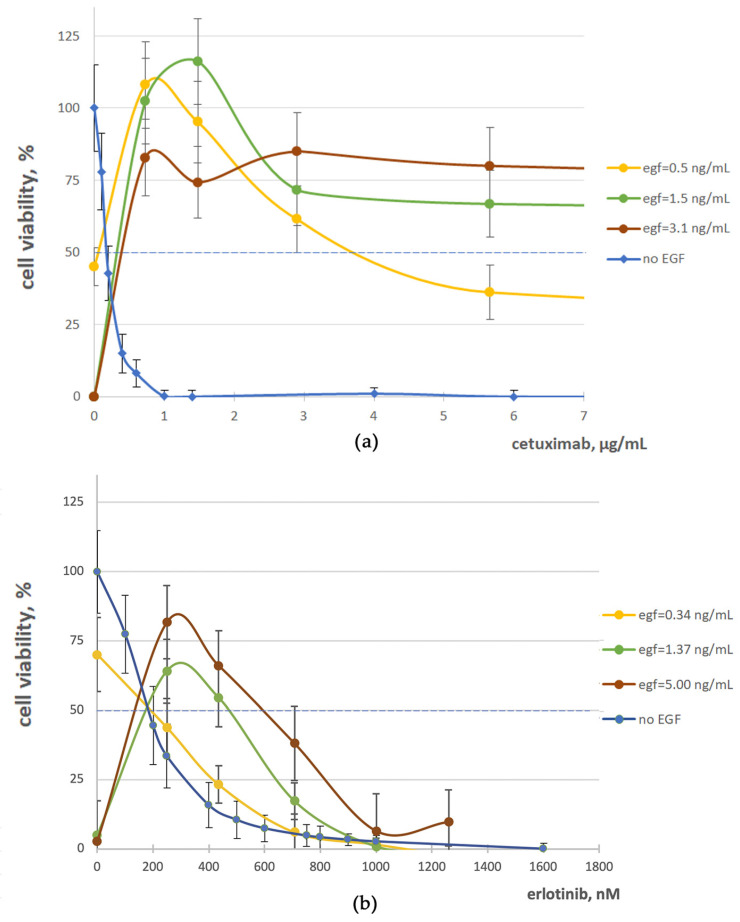
A431 cell growth rate with cetuximab (**a**) or erlotinib (**b**) without EGF (blue curve) and in the presence of EGF (yellow, green, and red curves), in different concentrations. Dose–response curves were calculated using at least three biological replicates of every experiment normalized to no-drug conditions with FBS-only growth media (FBS).

**Figure 5 cells-12-02022-f005:**
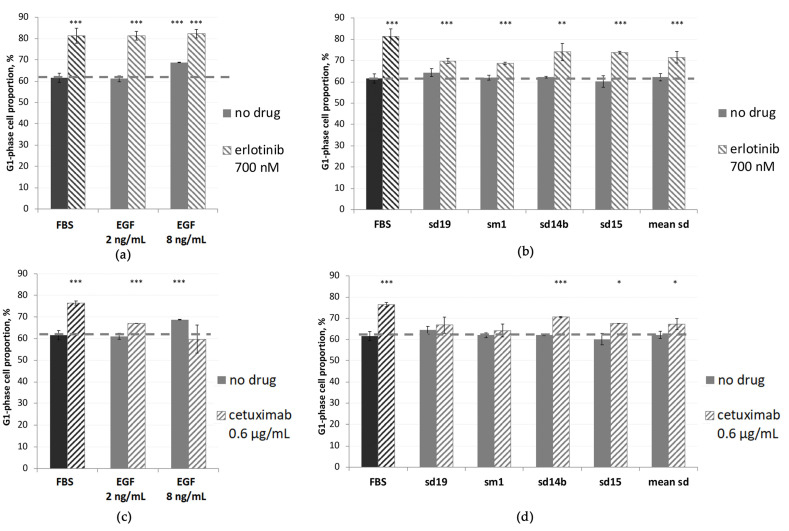
The proportions of A431 cells in G1 phase of cell cycle after treatment with erlotinib (**a**,**b**) and cetuximab (**c**,**d**) in the presence of EGF or 5% human serum samples compared with non-treated cells (“FBS” black bars); Asterisks stand for statistically significant differences between FBS and other samples: *, *p* < 0.05; **, *p* < 0.01; ***, *p* < 0.001. The cell cycle distribution analysis was carried out by FACS analysis after cell staining with propidium iodide. Each bar represents the mean ± S.D. of three independent experiments.

**Figure 6 cells-12-02022-f006:**
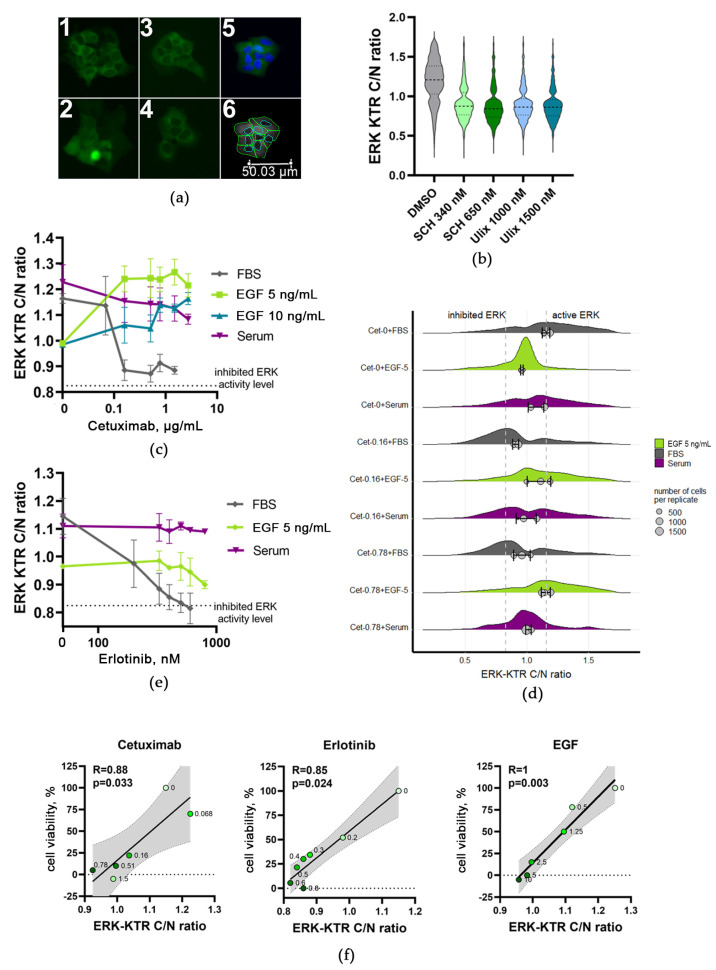
A431 ERK KTR values after treatment with cetuximab (indicated in µg/mL), in the absence of EGF (FBS) and EGF at 5 and 10 ng/mL. (**a**) Microscopy images of ERK-KTR reporter in A431 cells treated for 24 h with DMSO (1), cetuximab (2), blood serum (3), and cetuximab in combination with blood serum (4). To segment cell nuclei, they were stained with Hoechst-33342 1 h before imaging (5); cytoplasm and nuclei segmentation were performed using CellProfiler (6). (**b**) Violin plots showing ERK KTR values distribution for A431 cells treated with selective ERK inhibitors SCH772984 (SCH) and ulixertinib (Ulix). (**c**) Curves represent the median ERK KTR for each condition (cetuximab concentrations are indicated) calculated from three replicates. ERK-KTR intensity fluorescence ratio in cytoplasm to nucleus (C/N ratio) corresponds to ERK activity levels. The dotted line shows the ERK activity level in A431 treated with selective ERK1/2 inhibitors. For each data point, the SD is shown. (**d**) ERK activity distribution when A431 cells were treated with cetuximab alone or in the presence of 5 ng/mL EGF. For each condition, the ERK activity was measured in 300–1000 individual cells for three biological repeats. (**e**) Curves represent the median ERK KTR for each condition (erlotinib concentrations are indicated) calculated from three replicates. The ERK-KTR intensity fluorescence ratio in cytoplasm to nucleus (C/N ratio) corresponds to ERK activity levels. The dotted line shows the ERK activity level in A431 treated with selective ERK1/2 inhibitors. For each data point, the SD is shown. (**f**) Spearman correlation between ERK activity (ERK-KTR C/N ratios) and cell growth for cetuximab, erlotinib, and EGF. Concentrations of cetuximab (µg/mL), erlotinib (µM), and EGF (ng/mL) are marked for each data point. The line shows linear regression, and the gray area shows confidence interval.

**Figure 7 cells-12-02022-f007:**
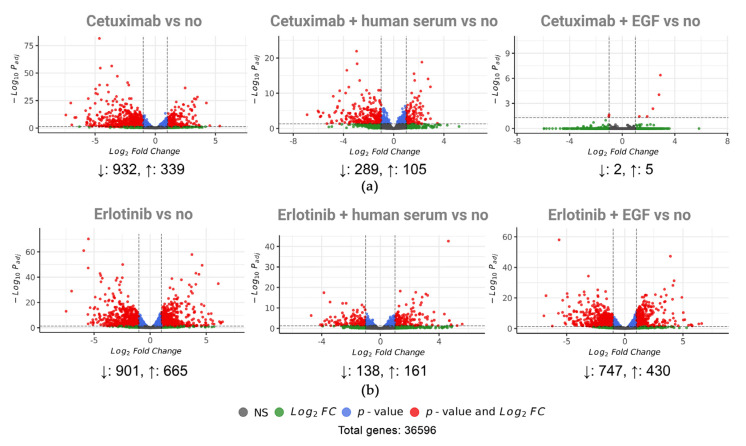
Differentially expressed genes (DEGs) in comparison with no-drug conditions for (**a**) cetuximab; cetuximab and human serum; cetuximab and EGF; and for (**b**) erlotinib; erlotinib and human serum; erlotinib and EGF. DEGs are shown in red (log2FC > 1 or log2FC < −1, adjusted *p*-value < 0.05).

**Figure 8 cells-12-02022-f008:**
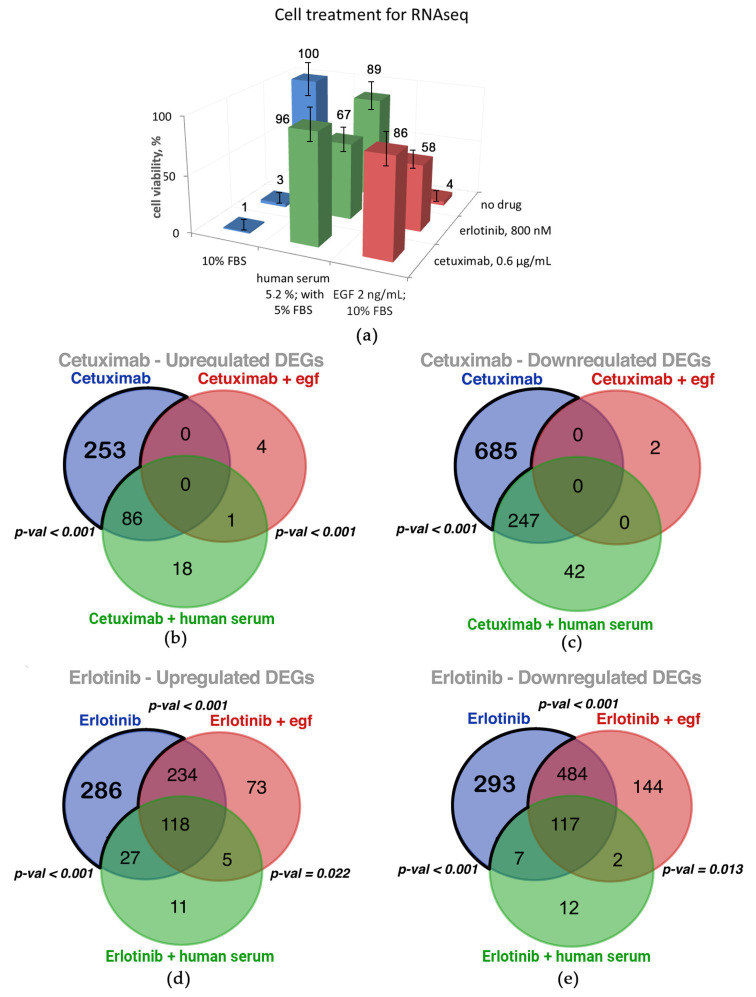
(**a**) Cell growth under the cell treatment conditions used to collect samples for RNA sequencing. (**b**–**e**) Venn diagram showing the overlap in differentially expressed genes (DEGs) in the presence of cetuximab only and cetuximab along with human serum or along with EGF, (**b**,**c**); and in the presence of erlotinib, (**d**,**e**). Cetuximab core and Erlotinib core gene groups are outlined.

**Figure 9 cells-12-02022-f009:**
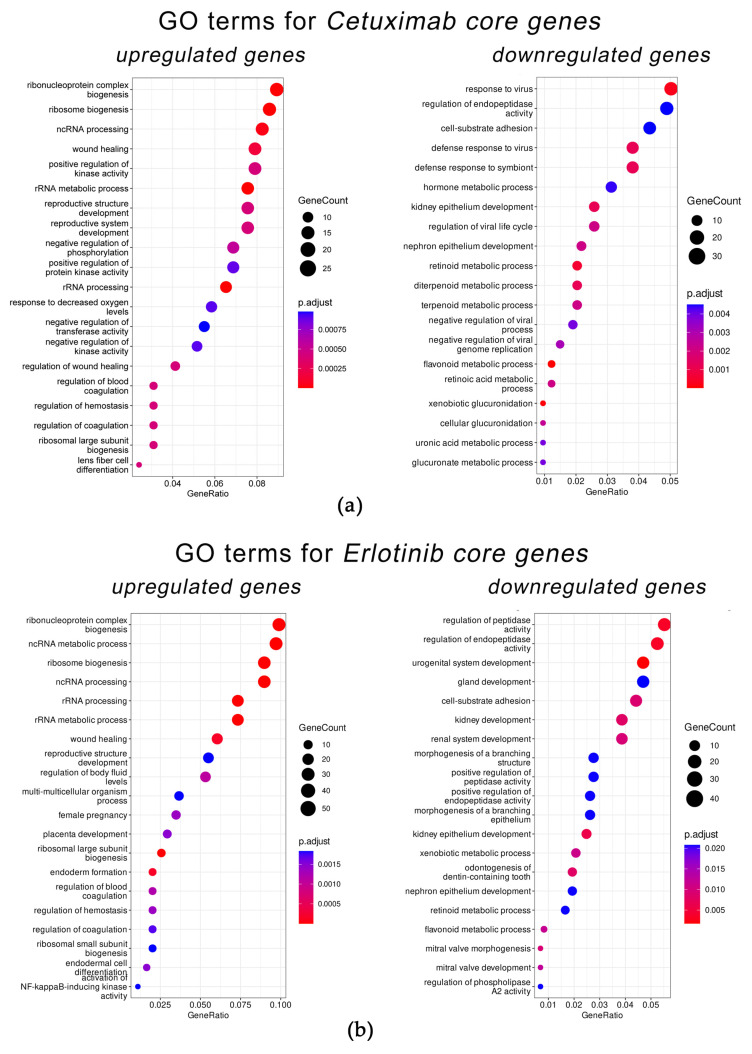
Top 20 Gene Ontology (GO) terms enriched among upregulated and downregulated genes in the Cetuximab core gene group (**a**) and in the Erlotinib core gene group (**b**). Visualized using R package enrichplot (http://bioconductor.org/packages/release/bioc/html/enrichplot.html (accessed on 15 June 2023)). All terms passed the Benjamini–Hochberg adjusted *p*-value threshold of 0.05.

**Figure 10 cells-12-02022-f010:**
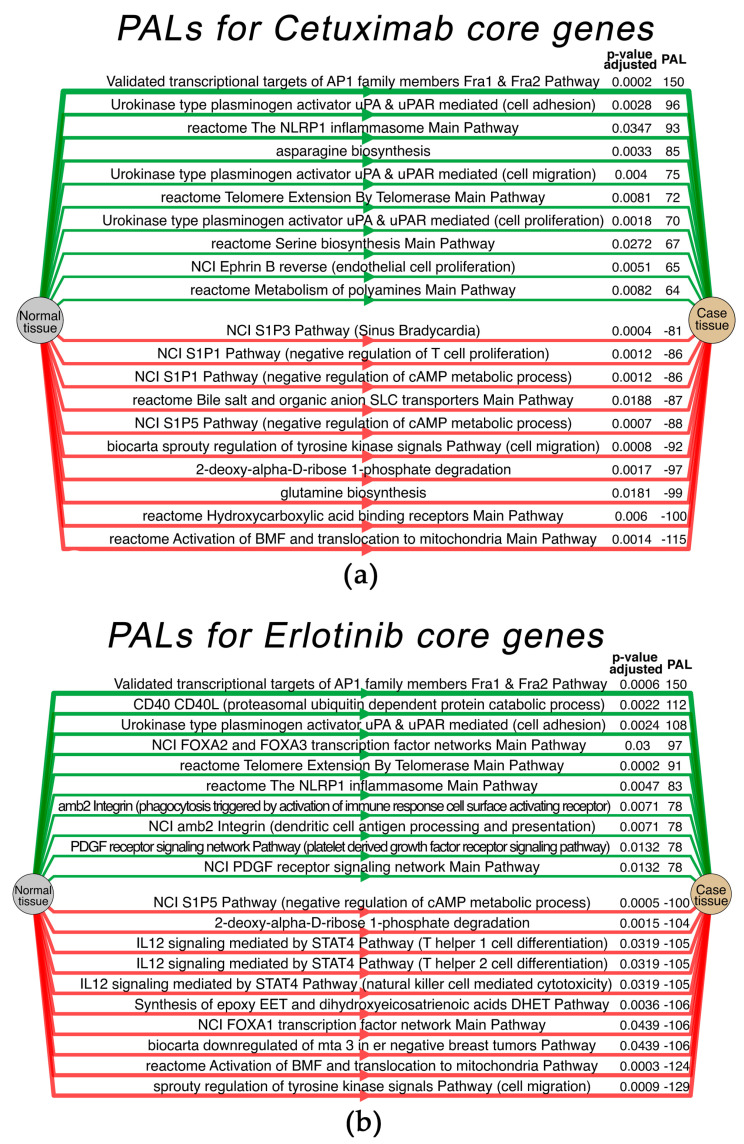
Top ten most strongly activated and inhibited molecular pathways calculated for the Cetuximab core gene group (**a**) and the Erlotinib core gene group (**b**) in A431 cells. PAL value and adjusted *p*-value are shown.

**Figure 11 cells-12-02022-f011:**
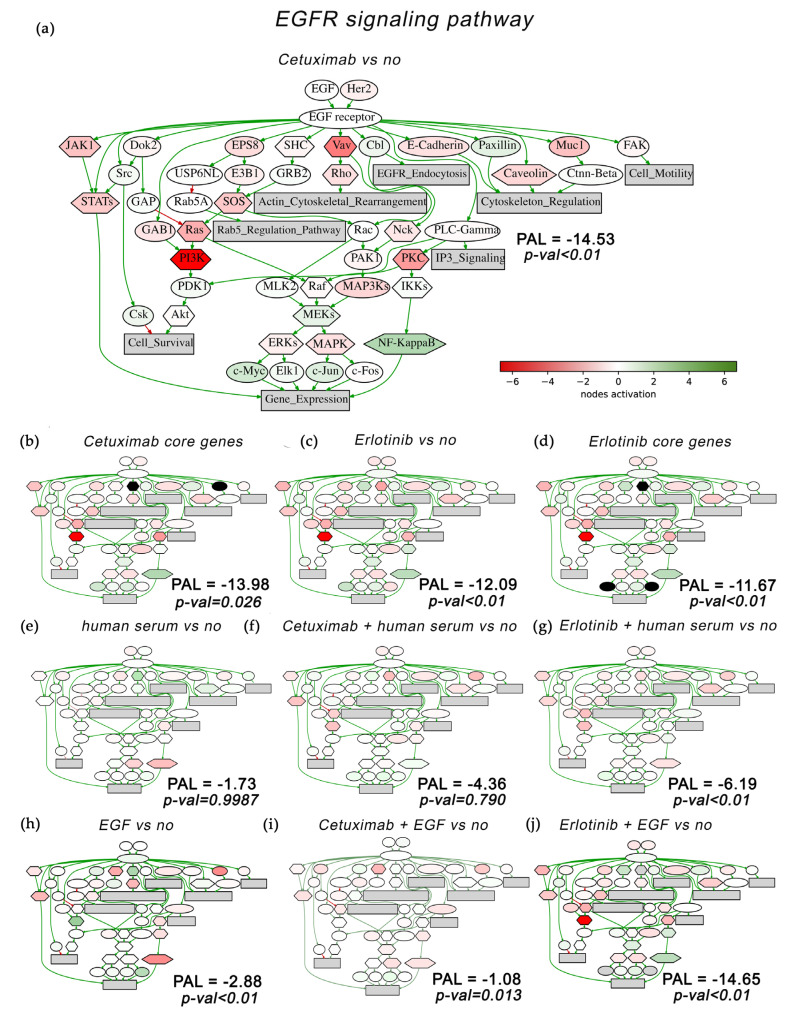
Activation chart of EGFR pathway (Qiagen Pathway Central) shown as an interacting network. EGFR pathway components activation compared with no-drug conditions is shown for: (**a**) Cetuximab-only treatment; (**b**) Cetuximab core gene group; (**c**) Erlotinib-only treatment; (**d**) Erlotinib core gene group. EGFR pathway activation in the presence of human blood serum only treatment (**e**), in combination with cetuximab (**f**), and in combination with erlotinib (**g**) are shown. EGFR pathway activation in the presence of EGF only treatment (**h**), in combination with cetuximab (**i**), and in combination with erlotinib (**j**) are shown. EGFR pathway activation level (PAL) is indicated for each gene group. Green/red arrows indicate activation/inhibition interactions, respectively. The color depth of transcript nodes reflects the extent of node activation (natural logarithms of the expression fold change for each node; the reference is the geometric average between expression levels in all samples in the respective groups). Green stands for activation, red stands for inhibition, white stands for non-differential expression. On (**b**,**d**), black ovals show nodes for which the activation caused by cetuximab-only treatment differs from the cetuximab core gene group or the activation caused by erlotinib-only treatment differs from the erlotinib core gene group. Visualized using Oncobox software.

**Figure 12 cells-12-02022-f012:**
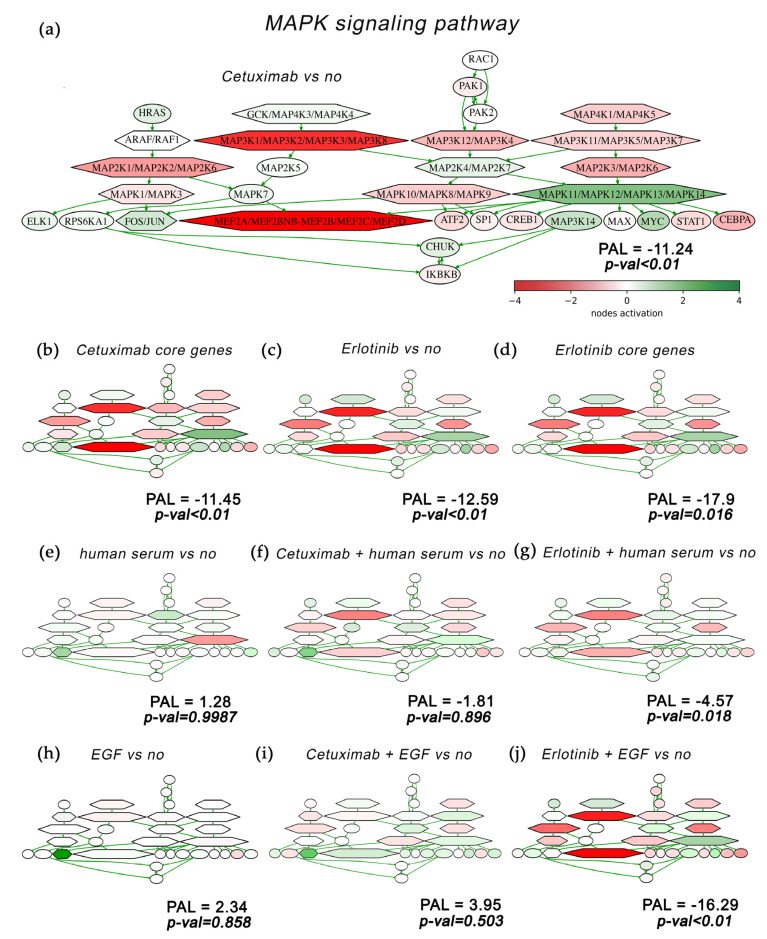
Activation chart of MAPK pathway (Biocarta) shown as an interacting network. MAPK pathway components activation compared with no-drug conditions are shown for: (**a**) Cetuximab-only treatment; (**b**) Cetuximab core gene group; (**c**) Erlotinib-only treatment; (**d**) Erlotinib core gene group. MAPK pathway activation in the presence of human blood serum only treatment, (**e**); in combination with cetuximab, (**f**); and in combination with erlotinib, (**g**), are shown. MAPK pathway activation in the presence of EGF-only treatment, (**h**); in combination with cetuximab, (**i**); and in combination with erlotinib, (**j**), are shown. MAPK pathway activation in the presence of EGF-only treatment and in combination with cetuximab or erlotinib is shown in panel F. MAPK pathway activation level (PAL) is indicated for each gene group. Green/red arrows indicate activation/inhibition interactions, respectively. The color depth of transcript nodes reflects the extent of node activation (natural logarithms of the expression fold change for each node, the reference is the geometric average between expression levels in all samples in the respective groups). Green stands for activation, red stands for inhibition, white stands for non-differential expression.

## Data Availability

RNA sequencing data are available through the link https://www.ncbi.nlm.nih.gov/bioproject/PRJNA966187 other data included in the manuscript and Appendix A.

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
