# Peer review of "Human Blood Serum Can Diminish EGFR-Targeted Inhibition of Squamous Carcinoma Cell Growth through Reactivation of MAPK and EGFR Pathways"

_cells, 2023, doi:10.3390/cells12162022_

Round 1

Reviewer 1 Report

Kamashev et. al. submitted their original research article entitled “Human blood serum can diminish EGFR-targeted inhibition of squamous carcinoma cell growth through reactivation of MAPK and EGFR pathways” to the Cells.

In the article, the authors discussed the abrogating role of human serum on the cell proliferation-hindering effect of EGFR blockers in squamous carcinoma cells. In general, the configuration and writing of the article have been adequately implemented with minor grammar errors. The authors provided comprehensive background information regarding the therapeutical application regulations of EGFR inhibitors and the intracellular downstream modulations following the treatment. They also addressed several obscurities in the activation mechanism of erlotinib and cetuximab and why certain patient groups fail to respond to the treatment appropriately. Therefore, I suggest a minor revision for this original research article to be considered for publication in the Cells. I listed my comments below for the improvement of the manuscript:

Comment #1: The abbreviations used throughout the article, including differentially expressed genes (DEG) and gene ontology (GO), are repeated many times. I suggest either removing the abbreviations altogether and using the actual term or explaining the abbreviation where it appears in the text for the first time only. 

Comment #2: I suggest replacing the term “cell growth” with either “cell proliferation” or “cell viability” in the figures and the manuscript since readers may mistakenly interpret cell growth as the size of the cells rather than their multiplication capacity.

Comment #3: It was discussed in the manuscript that human serum restored cell proliferation and cell cycle progression in cells treated with cetuximab and erlotinib. The authors should provide information about the components (TGF-β, amphiregulin, epiregulin, etc.) in the serum that interfere with the therapeutic impact of the EGFR blockers. Is it EGF alone, and if it is, what is the average physiological concentration of EGF in human serum? Is the EGF level or EGFR expression altered in patients with squamous carcinoma?

Comment #4: In Figure 1, it is seen that A431 cells incubated with human serum samples have a lower percentage of cell viability (except for three samples, sd14, sd15, sd8) when compared to the cells treated with FBS. If human serum reduces cell viability, what is the underlying mechanism of the dual treatment of human serum and EGFR blockers restoring cell viability? Why human serum alone did not affect the ERK1/2 signaling pathway, but it could restore the activity by co-application with cetuximab or erlotinib? Page 23, Line 758 “……..In contrast, mixed human blood serum sample alone did not affect ERK1/2 activity, but it could almost completely restore ERK1/2 activity of A431 cells to no-drug level for both cetuximab and erlotinib”.

Comment #5: The authors stated in section “3.3. Dual effects of EGF on A431 cell growth and resistance to EGFR inhibitors” that lower concentrations of EGF promote cell proliferation while suppressing it in higher. The authors should comment on the possible reason(s) behind this phenomenon and why its effect is switched once co-treated with EGFR blockers.

Comment #6: In Figure 4, the bell-shaped cell proliferation curve presentation is very confusing. The authors should either pick 0 or 100% cell viability as a starting point for all the treatments or use ratios for comparison.

Comment #7: The authors analyzed pathways activated and inhibited following cetuximab and erlotinib treatment (Figure 9). They should discuss which common pathway(s) can be considered the most consequential for cell proliferation and cell cycle progression in the case of squamous carcinoma.

Comment #8: The authors should elaborate on the clinical point-of-view on the application of EGFR inhibitors (either monoclonal antibody or small molecule-based), how to design an effective treatment according to the level of EGFR ligands in patient blood, and if there should be a dose adjustment of the blockers to compete with the ligands. 

 In general, the configuration and writing of the article have been adequately implemented with minor grammar errors. 

Author Response

We thank reviewers for thorough analysis of the manuscript, useful suggestions and the questions which allow us to improve the manuscript.

Here is a point-by-point response to your comments:

Comment #1: The abbreviations used throughout the article, including differentially expressed genes (DEG) and gene ontology (GO), are repeated many times. I suggest either removing the abbreviations altogether and using the actual term or explaining the abbreviation where it appears in the text for the first time only. 

>> We agree

and replaced differentially expressed genes with abbreviation (DEGs). Although in figure legends we left differentially expressed genes as otherwise it could be difficult to understand figures without reading the text.  

Comment #2: I suggest replacing the term “cell growth” with either “cell proliferation” or “cell viability” in the figures and the manuscript since readers may mistakenly interpret cell growth as the size of the cells rather than their multiplication capacity.

>>We agree

and replaced this captions to cell viability

Comment #3: It was discussed in the manuscript that human serum restored cell proliferation and cell cycle progression in cells treated with cetuximab and erlotinib. The authors should provide information about the components (TGF-β, amphiregulin, epiregulin, etc.) in the serum that interfere with the therapeutic impact of the EGFR blockers. Is it EGF alone, and if it is, what is the average physiological concentration of EGF in human serum? Is the EGF level or EGFR expression altered in patients with squamous carcinoma?

>> We agree

 and added required information about HER ligand concentrations in section 3.3 as well as following sentence :

We measured EGF concentration in human sera used here by ELISA; it varied between 0.7 and 1.4 ng/ml. Thus, in our experiments, EGF concentration in the growth media supplemented with 5% human serum does not exceed 0.07 ng/ml. We, therefore, conclude that human blood serum influence on the erlotinib and cetuximab impact (Figures 2 and 3) cannot be explained by endogenous EGF action. EGF is a major HER ligand in human blood, and measured concentration of other HER ligands is much lower: TGF-alpha ~0.01 ng/ml; epiregulin ~0.3 ng/ml; betacellulin ~0.2 ng/ml (I added this information in the text).

Comment #4: In Figure 1, it is seen that A431 cells incubated with human serum samples have a lower percentage of cell viability (except for three samples, sd14, sd15, sd8) when compared to the cells treated with FBS. If human serum reduces cell viability, what is the underlying mechanism of the dual treatment of human serum and EGFR blockers restoring cell viability? Why human serum alone did not affect the ERK1/2 signaling pathway, but it could restore the activity by co-application with cetuximab or erlotinib? Page 23, Line 758 “……..In contrast, mixed human blood serum sample alone did not affect ERK1/2 activity, but it could almost completely restore ERK1/2 activity of A431 cells to no-drug level for both cetuximab and erlotinib”.

>>It is a very good question. We are intent to elaborate it in our further studies. We believe that one of the possible strategies to illuminate this question is following: we have to fractionate serum by several chromatography methods and to perform mass-spectrometric analysis of the active fractions to identify serum protein(s) responsible for the drug resistance and their targets in A431 cells.

Comment #5: The authors stated in section “3.3. Dual effects of EGF on A431 cell growth and resistance to EGFR inhibitors” that lower concentrations of EGF promote cell proliferation while suppressing it in higher. The authors should comment on the possible reason(s) behind this phenomenon and why its effect is switched once co-treated with EGFR blockers.

>> We comment the possible reason of EGF dual effects further in the text, in section 3.4, as it is related to cell cycle analysis, and we believe that cell growth rate is associated with cell cycle progression:

“This effect can be related to previously found EGF mediated upregulation of p21(Cip1) and cetuximab mediated upregulation of p27(Kip1)”

Comment #6: In Figure 4, the bell-shaped cell proliferation curve presentation is very confusing. The authors should either pick 0 or 100% cell viability as a starting point for all the treatments or use ratios for comparison

>>We agree

and made figure legend clearer. 

Figure 4. A431 cell growth rate with cetuximab (a) or erlotinib (b) without EGF (blue curve) and in the presence of EGF (yellow, green, and red curves), in different concentrations. Dose-response curves were calculated using at least three biological replicates of every experiment normalized to no-drug conditions with FBS-only growth media (FBS).

In Figure 4 we compare cell proliferation in the absence of EGF (blue curve which starts from 100%) with cell proliferation in the presence of fixed EGF concentration. At EGF concentration of 1.37 and 5 ng/ml, cell viability curves start from ~5% as EGF in the absence of erlotinib (or cetuximab) strongly inhibits cell proliferation.

Comment #7: The authors analyzed pathways activated and inhibited following cetuximab and erlotinib treatment (Figure 9). They should discuss which common pathway(s) can be considered the most consequential for cell proliferation and cell cycle progression in the case of squamous carcinoma.

>>This question is very important for understanding the mechanisms of drug action and drug resistance. Along with transcriptome profiling and bioinformatic analysis we also performed ERK activation measurements and cell cycle analysis. We pointed to MAPK pathway in discussion section. Here, in Results section, in Figure 9 (and 10) we listed GO terms and pathways most strongly activated and inhibited by the drugs calculated on the basis of transcriptome profiling by RNAseq.

Comment #8: The authors should elaborate on the clinical point-of-view on the application of EGFR inhibitors (either monoclonal antibody or small molecule-based), how to design an effective treatment according to the level of EGFR ligands in patient blood, and if there should be a dose adjustment of the blockers to compete with the ligands. 

>>As far as I know oncologists use maximal tolerated doses of targeted drugs. As we performed in vitro studies here, we could elaborate on the pre-clinical point-of-view only. We propose to check drug impacts in pre-clinical studies in the presence of human blood serum.

We added this point-of-view to the Discussion section

Reviewer 2 Report

In this study, the authors showed a significant  and donor-specific influence of human blood serum on the inhibition of cell growth rate by EGFR-targeted drugs, cetuximab and erlotinib. The experiment is systematically planned and the experimental data is convincing. For pathway analysis they employed high performance RNA sequencing to identify differentially expressed genes and  assess differential molecular pathway activation levels. This is a valuable study with translational value as a high proportion of patients are resistant to monoclonal antibodies that target the  extracellular domains of EGFR, and EGFR inhibitors.

There are only minor comments/ concerns that need to be addressed:

1   (1) There is no section on statistical analysis, so it is not clear what statistical methods were used to analyze the in vitro experimental data.

      (2) There were different responses observed depending on the specific donor serum used. Is there any correlation with donor characteristic such as age? After all, most cancers are prevalent in the older age population.

      (3) For the cell cycle analysis, it would be good to include the representative histograms of the different phases of the cell cycle with and without treatment  in the Supplementary information.

      (4) Figure 6a lacks a scale bar.

      (5) In the final paragraph of the “Discussion”, the authors may want to consider citing literature on the importance of personalized oncology. Would also like to  suggest one step for further testing of drug activity in the presence of blood  serum of individual patient and model cancer cell line, i.e. using same patient derived cultured cancer cells in place of a model cancer cell line.

Author Response

We thank reviewers for thorough analysis of the manuscript, useful suggestions and the questions which allow us to improve the manuscript.

Here is a point-by-point response to your comments:

(1) There is no section on statistical analysis, so it is not clear what statistical methods were used to analyze the in vitro experimental data.

>>We agree

 and we added section on statistical analysis to Materials and Methods.

2.9. Statistical analysis

Statistical analysis was performed using the GraphPad PRISM 6.0 software (GraphPad Software Inc.), and values of p < 0.05 were considered statistically significant. The data were represented as the mean ± SD of at least three experiments performed in different days.

To calculate IC50, the median-effect dose that inhibits the system under study by 50%, we used equestion: IC50 = D / (100/y -1)1/m, where D is the concentration of a drug; y, cell viability (%); and m is the coefficient signifying the shape of the dose-effect relationship.

Significance of pathway activation levels (PALs) were established using Oncobox pathway analysis method [60] for 1611 molecular pathways containing 10 or more gene products extracted from the public databases [58] using the original software [57]. For PAL calculations, each sample expression profile was normalized on mean geometrical levels of gene expression for all samples in the dataset under investigation.

We used Benjamini–Hochberg FDR-adjusted p-values < 0.05 as a cutoff value for filtering and selecting differentially expressed genes, pathways and GO terms.

To test whether a given number of common differential genes or pathways between the two of three intersecting datasets is significant, 1000 random intersections were performed according to [59]. In every case, two/three random samples from two/three corresponding gene sets of the respective datasets were taken. Then these random samples were intersected for each iteration and 1000 numbers of random common genes were obtained. p-value of intersection significance was calculated as a fraction of random numbers equal or higher than the experimentally observed number of common genes.

      (2) There were different responses observed depending on the specific donor serum used. Is there any correlation with donor characteristic such as age? After all, most cancers are prevalent in the older age population.

>>Unfortunately, the number of donors is not sufficient to explore statistically justified correlation between age and serum interference with the drug activities.

      (3) For the cell cycle analysis, it would be good to include the representative histograms of the different phases of the cell cycle with and without treatment  in the Supplementary information.

>>We agree.

We now added the required histograms to the Supplementary materials.

      (4) Figure 6a lacks a scale bar.

>>Thank you for noticing this. Corrected.

      (5) In the final paragraph of the “Discussion”, the authors may want to consider citing literature on the importance of personalized oncology. Would also like to  suggest one step for further testing of drug activity in the presence of blood  serum of individual patient and model cancer cell line, i.e. using same patient derived cultured cancer cells in place of a model cancer cell line.

>> Here, we performed in vitro studies. Thus, we could elaborate on the pre-clinical point-of-view only. We propose to check drug impacts in pre-clinical studies in the presence of human blood serum. We added this point-of-view in Discussion section.